# Progressive Evolutionary Dynamics of Gene-Specific ω Led to the Emergence of Novel SARS-CoV-2 Strains Having Super-Infectivity and Virulence with Vaccine Neutralization

**DOI:** 10.3390/ijms25126306

**Published:** 2024-06-07

**Authors:** Amit K. Maiti

**Affiliations:** Department of Genetics and Genomics, Mydnavar, 28475 Greenfield Rd, Southfield, MI 48076, USA; akmit123@yahoo.com or amit.maiti@mydnavar.com; Tel.: +1-248-379-3129

**Keywords:** SARS-CoV-2, synonymous, nonsynonymous, dn/ds, virulence

## Abstract

An estimation of the proportion of nonsynonymous to synonymous mutation (dn/ds, ω) of the SARS-CoV-2 genome would indicate the evolutionary dynamics necessary to evolve into novel strains with increased infection, virulence, and vaccine neutralization. A temporal estimation of ω of the whole genome, and all twenty-nine SARS-CoV-2 genes of major virulent strains of alpha, delta and omicron demonstrates that the SARS-CoV-2 genome originally emerged (ω ~ 0.04) with a strong purifying selection (ω < 1) and reached (ω ~ 0.85) in omicron towards diversifying selection (ω > 1). A marked increase in the ω occurred in the spike gene from alpha (ω = 0.2) to omicron (ω = 1.97). The ω of the replication machinery genes including *RDRP*, *NSP3*, *NSP4*, *NSP7*, *NSP8*, *NSP10*, *NSP13*, *NSP14*, and *ORF9* are markedly increased, indicating that these genes/proteins are yet to be evolutionary stabilized and are contributing to the evolution of novel virulent strains. The delta-specific maximum increase in ω in the immunomodulatory genes of *NSP8*, *NSP10*, *NSP16*, *ORF4*, *ORF5*, *ORF6*, *ORF7A*, and *ORF8* compared to alpha or omicron indicates delta-specific vulnerabilities for severe COVID-19 related hospitalization and death. The maximum values of ω are observed for spike (*S*), *NSP4*, *ORF8* and *NSP15*, which indicates that the gene-specific temporal estimation of ω identifies specific genes for its super-infectivity and virulency that could be targeted for drug development.

## 1. Introduction

The SARS-CoV-2 genome, like any other viral genome, continuously accumulates mutations by replicating mistakes and retaining the favorable mutations to achieve optimum functions for its proteins to evade the host immune system. This continuous accumulation of functional (nonsynonymous) mutations creates an evolutionary stabilized, or saturated virus for maximum adaptation with the host systems. The proportion (d_n_/d_s_, ω) of nonsynonymous (d_n_) to the synonymous mutation (d_s_) over time indicates the level of stringency of the host immune system that the virus is encountering [1]. In the neutral theory of virus evolution, the rate of synonymous mutation is greater than nonsynonymous mutation in a less constrained host environment, leading to purifying selection (ω < 1) [2,3] (Figure 1a). When stabilized in an ideal host environment, the accumulation of an equal proportion of synonymous and nonsynonymous mutations is expected, whereas in the stringent host environment, the virus accumulates more nonsynonymous mutations to adapt to the host immune system, leading to diversifying selection (ω > 1) [4,5]. A virus undergoes divergent selection when novel conditions are created, due to environmental changes over a short period or a heterogeneous environment with multiple niches [6]. SARS-CoV-2 has a substitution rate of 2.22 nt/month with an evolutionary rate of 9 × 10^−4^/site/year [7,8]. SARS-CoV-2 evolution from bat RaTG13 strictly follows neutral evolution with strong purifying selection, whereas its propagation in the human proceeds towards divergent selection due to its ability to infect multiple human organs that act as constrained environments because of variations in immune surveillance [7]. This property enables SARS-CoV-2 to select more nonsynonymous mutations to survive aggressively, and to become virulent.

Capturing the evolutionary dynamics of SARS-CoV-2 leading to the emergence of novel virulent strains is challenging, partly due to its differential rate of transmission in various parts of the world. Furthermore, inequal incorporation, different types of vaccine use, and the social interconnectedness of the vaccine mandate complicated the selection of favorable mutations. Studying the SARS-CoV-2 genome in a particular area or country for evolutionary perspectives is not sufficient to comprehend its favorable mutation selection. The ω at a particular time point in the evolutionary path of the virus indicates its adaptation status against the host immune systems. The temporal change in ω in each gene is an evolutionary measure of change in overall gene/protein sequences that contributes to better survival, leading to the emergence of a novel virus with a higher infecting ability and virulence. Thus, the estimation of ω would provide the weakness or strength of the gene/protein challenging the host immune systems and help to design efficient vaccines and therapeutic drugs against the virus.

The infective power of SARS-CoV-2 depends on the S (spike) protein that has seven entry-point residues at the RBM (Receptor Binding Motif) to attach with the human ACE2 receptor [9]. Among these residues, the K353-501N interactions almost gave a passport to the virus to enter the human host from bat RATG13 lineage [7]. A spike mutation, D614G, is more vulnerable to people having an SNP rs35074065 in cis-eQTL with MX1, TMPRSS2 gene [7], and D614G indeed contributed in increased infectivity [10]. All super-infective strains including alpha (B.1.1.7), delta (B.1.6.1.7), and omicron (BA series) possessed a D614G mutation, but had a different level of infective power, and vaccine neutralizing abilities that were due to other mutations in the spike gene [11]. Omicron had the highest transmissibility compared to alpha and delta, creating the maximum wave of infections worldwide [12,13]. Similarly, virulency in these three major strains showed various levels of severity leading to COVID-19 hospitalization and related death [14,15,16,17]. It is observed that the delta variant is more severe than other two major strains, alpha and omicron, in respect to increased hospitalizations and mortalities [18,19]. Virulency depends on the severity of symptoms when a virus suppresses, modulates, and evades the immune system by engaging its proteins to act as antagonists against host immunomodulatory surveillance proteins. SARS-CoV-2 protein-specific induction involves the inhibition of interferon (IFN) pathways, C3 activation, and dysfunctional inflammasome formation leading to a “cytokine storm”, and multiple organ damage [19,20].

I estimated the ω of SARS-CoV-2 whole genome (WG) in various parts of the world and its twenty-nine genes in its major strains of alpha, delta, and omicron to elucidate how the accumulation or reduction of nonsynonymous mutations contributes to its infective ability, virulence and vaccine neutralization. Most of the genes, including spike- and replication-associated genes, show an increased ω in omicron, and immunomodulatory genes show the highest ω in delta, correlating with the delta-specific highest vulnerability, that subsequently reduced in omicron. The maximum increase in ω occurred in NSP4, ORF8, NSP15 and the spike gene with a profound effect on infectivity, virulence and vaccine neutralization.

## 2. Results

### 2.1. Cladistic Phylogenetic Differences of Major Strains of SARS-CoV-2

Since its emergence in humans, the SARS-CoV-2 has accumulated mutations to create genetic distances, leading to divergence into novel strains. A typical alpha, delta, or omicron genome differs from the original strain on average by 22.7 nt, 48.5 nt, and 74.1 nt, respectively (Appendix A). Whereas the number of mutations accumulated in a diverse population of approximately 100 genomes of each strain is equivalent to 131 nt (alpha), 475.5 nt (delta), and 301.25 nt (omicron). Omicron originated with considerable genetic distances, with a less [21] total number of mutations from their nearest ancestor delta (Figure 1c,d). This reduction is attributed to the bias of the host RNA editing enzymes APOEBAC and ADAR, which function differently for omicron than delta [22]. Similarly, among only omicrons, large differences in nt replacements in a month period between January 2022 (365 nt), March 2022 (326 nt), April 2022 (267 nt), and May 2022 (247 nt) made these strains cladistically separable (Figure 1b) and justify the subdivision of omicron to a later series of isolates, such as BA.1, BA.2, BA.3, BA.4, etc. [23].

### 2.2. The ω of the Whole SARS-CoV-2 Genome Is Increased since Its Emergence with an Approach to Diversifying Selection

When SARS-CoV-2 entered the human host from bats, a new immuno-environmental challenge required modifications of its own protein functions to counter the host immune surveillance. An increasing ω (dn/ds) signifies the accumulation of more nonsynonymous mutations over synonymous mutations, which are missense mutations that change amino acids into their corresponding protein sequences that shape the genes/proteins for achieving optimum function to evade human immune systems. The overall changes of ω in the genome represent directions of positive or negative selections, and the measure of its evolutionary status in encountering the host immune systems.

The ω of the WG for the conversion of bat coronavirus RATG13 lineage to SARS-CoV-2 is approximately 0.04 [7] with purifying selection, indicating that a minimal nt replacement in its genome allowed it to enter into a human host. Upon propagation in humans, the ω of the WG increased to alpha (0.30–0.35, September 2020), delta (ω = 0.56, June 2021) and omicron (ω = 0.79–0.85, January 2022) (Figure 2a,b, Table 1), which enabled SARS-CoV-2 evolution to proceed towards diversifying selection (ω > 1). A little decline is observed in the later omicrons of March (ω = 0.64), April (ω = 0.68), and May in 2022 (ω = 0.67), implying most genes started becoming stabilized in the human immune environment.

Despite the estimation of the mutation rate of SARS-CoV-2 (9 × 10^−4^/site/year) [7,8], the rate of nonsynonymous over synonymous mutation, which actually impacts the viral functions, has not been estimated earlier. I estimated that the instant rate of change in the ω (dω/dt) at any given time is 0.03/month (1.2 × 10^−5^/site/year) and the strain-specific average rate of change in the ω is 0.025/month (1.00 × 10^−5^/site/year) (Figure 2c, Appendix A).

### 2.3. The Increase in the ω of Spike (S) Gene Increases Infection

As the infection abilities of major SARS-CoV-2 strains are different [24], we evaluated the evolutionary changes in the spike gene that are instrumental to attaching to the ACE2 receptor. In addition, after virus–host cell post-fusion, the S2 protein with N-glycan protects the virus by inducing non-neutralizing antibody responses [25]. The continuous accumulation of nonsynonymous mutations in spike gene from alpha (ω = 0.15) to early omicron (ω = 1.97, January 2022) (Figure 2d, Table 1) is justified later with the highest number of infections in the world during this period. A further increase in ω in the following months in March 2022 (ω = 2.4) supports the continuous refinement of spike protein. After a little decline in April 2022 (ω = 1.07), the ω again increased in May 2022 (ω = 1.43), suggesting that the spike attachment mechanism is constantly evolving to maximize its infection.

It is particularly notable that the rate of instant change in the ω of the spike gene in omicron is higher (3.2 × 10^−5^/site/year, January 2022) than delta (2.7 × 10^−5^/site/year, June 2021) or alpha (6.7 × 10^−6^/site/year, September 2020) (Figure 2e, Appendix A). The rate of the ω is markedly increased in omicron over a shorter time (5 months, July 2021–January 2022) than the evolution of delta from alpha (9 months, September 2020–June 2021) or alpha from the original strain (9 months, January 2020–September 2020). This is attributed to the increased infection power of omicron that faced various the environmental niches (genetic, ethnic, socio-economic variability of humans all over the world and the pre-existence of other diseases that made the host immune challenges variable) of human beings. Omicrons’ widespread infectivity used to overcome host immune challenges presumably due to the attack of multiple human organs where the ACE2 expression is high enough [26]. Indeed, the average rate of change in the ω of spike gene is much higher in omicron (7.2 × 10^4^/site/year) than in delta (4.9 × 10^−5^/site/year) or alpha (5.8 × 10^−6^/site/year). Thus, the increased infection power is directly proportional to the accumulation of nonsynonymous mutations in the spike gene.

### 2.4. The ω of the Other Viral Entry Associated Genes Are Continuously Increasing

The leader protein NSP1 binds with the 40s ribosome and degrades host mRNA, but keeps the viral mRNA intact with a potent inhibitor function of host gene expression that confers immunity [27]. NSP1 inhibits the RIG1-mediated sensing of the virus and suppresses the innate immune response [28]. An increased ω from alpha (ω = 0.0) to delta (ω = 0.83) to omicron (ω = 1.96, April 2022) (Figure 2f, Table 1) indicates its constant evolution by accumulating nonsynonymous mutations. Mou et al. (2021) identified 933 nonsynonymous mutations in *NSP1* and the majority of them exhibited a gain in flexibilities of the protein function leading to increased pathogenicity in delta [29] and S135R, K141-, S142-, and F143 mutation showed even higher pathogenicity in omicron [30,31].

Similarly, endonuclease NSP15 specifically degrades the viral polyuridine sequences to prevent the detection of the virus by the host immune sensor proteins, IFIH1, OAS2, OAS3, and RnaseL [32,33]. This induces tracheal epithelial damage, mucosal injury, and ciliary shedding with severe lymphocyte infiltration [34]. The T112I mutation in NSP15 in omicron indicated an uncontrolled evasion of the virus in a short period of time [35]. The ω of *NSP15* from alpha (ω = 1.07) was increased maximally in earlier omicron variants (ω = 3.56) (Figure 2g, Table 1) and contributes to some of the omicron-specific elevated immune escape compared to alpha or delta.

### 2.5. The ω of Replication and Multiplication Machinery Assisted Genes Are Markedly Increased

SARS-CoV-2 replication machineries (Appendix A) are constantly evolving for efficient functions for its multiplication. An increased ω is observed for RDRP (RNA Dependent RNA Polymerase, *NSP12*) since its emergence from the original strain to alpha (ω = 0.27), to delta (ω = 0.68), and to omicron (ω = 0.72, January 2022) (Figure 3a, Table 1). *RDRP* is a mutational hotspot and most mutations in this gene, results in decreased proofreading activities, leading to the emergence of novel SARS-CoV-2 variants [31]. NSP8 complexes with RDRP and A14S/V, A27T/V, and VI156L mutations increase the RDRP–NSP8–NSP7 complex formation [36]. The highest ω of *NSP8* in omicron (ω = 0.76) over alpha (ω = 0.0) or delta (ω = 0.16) justifies its increased replication [37] (Figure 3b, Table 1). NSP7, a NSP7–NSP8–RDRP replication complex protein [38], also has an increased ω in omicron (ω = 0.5) compared to the alpha (ω = 0.00) (Figure 3c, Table 1). The S25L and S26F in NSP7, along with P323L mutation in RDRP, facilitates the more efficient replication complex formation in omicron [39]. The NSP9 binds with its own viral ssRNA to stabilize it during viral replication and inhibits the host cyclin-CDK complex, resulting in hyperphosphorylation and leading to the attenuation of host replication [40]. An increased ω from the alpha (ω = 0.00) to omicron (ω = 0.4, April 2022) favored the better of these functions (Figure 3d, Table 1). Bansal et al. (2022) identified nine mutations in omicron that are implicated in the higher interaction of NSP9 with RDRP, leading to an increased replication of the virus [41]. Another viral replication-assisting protein, NSP3, consists of eight domains, and among them a papain-like domain provides a cleavage site for ubiquitination, and cleaves post-translated host proteins, a mechanism for evading host immune responses [42]. The X domain (MAC1) is indispensable for RNA replication. The increased ω of *NSP3* from alpha (ω = 0.2) to omicron (ω = 0.8) suggests that *NSP3* is approaching diversifying selection for its highest functional efficiency (Figure 3e, Table 1). A series of mutations in omicron (K38R, S1265, L1266I, A1892T, T24IG489S) in NSP3 affect the binding with ER, ssRNA and nucleocapsid to increase immune evasion with higher hospitalization [30].

Similarly, the NSP3 interacting protein, NSP4, is essential for viral replication [43], and interacts with the mitochondrial proteins TIMM9, TIMM10, TIMM10B, and TIMM29 [44]. Several NSP4 mutations in omicron (T492I, L264F, T327I, L438F) are implicated in the increased Double Membrane Vesicle (DMV) formation that are required for viral replication and budding [30]. The striking increase in the ω of *NSP4* from alpha (ω = 0.00) to delta (ω = 0.63) to omicron (ω = 4.94) (Figure 3f, Table 1) indicates its exceptional changes to achieve its optimal function. The marked increase in the ω of the ORF9 Nucleocapsid (N) protein (ORF9A + ORF9B) in omicron (ω = 1.47) from alpha (ω = 0.79) (Figure 3g, Table 1) is instrumental in providing an upgradation of the SARS-CoV-2 replication system and host immune evasion. Two mutations, R203K/G204R, in consecutive bases in ORF9, are enough to increase *ORF9* expression, leading to higher replication efficiency and excess host immune suppression in alpha than in the original strain [45,46]. ORF9 mutations P10S, E27-, N28-, and A29- in omicron exhibit IFN antagonism and increase replication, pathogenicity, and fitness by modulating host–virus interactions compared to other variants [12]. Helicase NSP13 introduces the 5′-cap of the viral mRNA with a suppression of IFN signaling by perturbing JAK1 phosphorylation in the STAT1 signaling pathway [47]. The R392C mutation in omicron is implicated in increased helicase activities in dsRNA unwinding, with a higher transmission rate and immune evasion [30]. The ω of *NSP13* is increased in the initial phase of omicron (ω = 1.06) but is decreased in alpha (ω = 0.18) and delta (ω = 0.6) (Figure 3h, Table 1), implying better replication efficiency in omicron than in the other two strains.

### 2.6. The ω of Virus Assembly and Release Proteins Are Increased

ORF4 (E) is confined to ER, Golgi complex, the ER–Golgi Intermediate Compartment (ERGIC), and involved in the assembly of virus particles and budding. It acts as an ion channel, specifically like viroporin, which enhances membrane permeability [48]. *ORF4* improves from alpha (ω = 0.00) to omicron (ω = 1.15) for optimum function (Figure 4a, Table 1). The T9I mutation in omicron provides stronger anchoring to the viral membrane and more pathogenicity over alpha and delta [31,49]. NSP14 (N7-methyltransferase) interacts with IMPDH2 to suppress numerous host gene expressions [50], and I42V mutation in omicron affects the proofreading of newly synthesized viral RNA and inhibits interferon signaling [30,31]. It is continuously evolving with an increased ω in omicron (ω = 1.33) from alpha (ω = 0.33) (Figure 4b, Table 1).

ORF3A, an integral membrane viroporin, acts as an ion channel to promote virus release [51]. The TRAF3-binding motif in ORF3A activates the NLRP3 inflammasome, stimulating pro–IL-1β and Il18 to promote the generation of a “cytokine storm” [20]. NLRP3 activation essentially regulates severe organ injury [52]. In omicron, G26144T mutation results in disorganized protein–protein interactions with TRAF3 promoting inflammation, and T223I leads to increase in “cytokine storm” [30,53]. The marked increase in the ω in early omicron (ω = 3.39) (Figure 4c, Table 1) than alpha (ω = 1.85) or delta (ω = 1.69) explains the early omicron-specific symptomatic differences for immune resistance. The transmembrane protein ORF7B resides in the Golgi complex and is implicated in epithelial cell adhesion, leading to heart rate dysregulation [54].

The L11F mutation in ORF7B in omicron resulted in signal modulation and IFN antagonism [30,53] over delta and alpha. An increase in the ω from the alpha (ω = 0.00) to the omicron (ω = 1.31) signifies symptomatic differences in omicron from alpha and delta (Figure 4d, Table 1).

### 2.7. Genes with Unchanged or Decreased ω Implicate Well Adaptation in Human Host with Efficient Functions

When environmental constraints are not present, the accumulation of synonymous over nonsynonymous mutation decreases ω in a gene [4,6], and an unchanged ω indicates that its optimal function was already achieved and does not need further refinement. NSP6 prevents the autophagosome formation from delivering viral components for degradation in lysosomes. The ω of NSP6 are almost unchanged from alpha (ω = 0.40) to omicron (ω = 0.55, May 2022) (Figure 4e, Table 1). The mutations L105* and S106* (* = any base) in omicron, and G107* in alpha, do not alter NSP6 global structure and function [30]. ORF10 alters binding affinity to respective HLA alleles, and predominantly decreases the affinity of epitopes to escape the host immune system [55]. ORF10 is a highly conserved protein and is not essential in humans [56]. Supporting this view, no major changes in the ω (Figure 4f, Table 1) are observed in *ORF10* from its emergence to omicron, indicating that this gene is acting at its highest level. NSP2 is involved in the endosome transport, translation, and modification of lipids [52]. The ω of *NSP2* is marginally decreased from alpha (ω = 0.47) to omicron (ω = 0.45) (Figure 4g, Table 1). No significant nonsynonymous mutation is identified in *NSP2* [30] and it is presumably functioning optimally in the human host.

### 2.8. The Increased Virulence of Delta Could Be Attributed to Increased ω in Immunomodulatory Genes

The delta-specific maximum increase in the ω, specifying the optimal function of these proteins, could be associated with the virulency differences of the other SARS-CoV-2 strains (Appendix A). ORF8 binds to the IRF3 to inactivate IFN induction and is critical to induce a “cytokine storm” by activating the IL17 pathway [57]. It also activates the complement protein C3 that contributes to the prothrombotic and proinflammatory states facilitating end-organ damage, including liver, kidney, heart, and brain, that are observed in severe cases of COVID-19 [19]. A natural variant of SARS-CoV-2 strain with 382 nt deletion in the *ORF8*gene in Singapore and Taiwan exhibited milder disease phenotypes, increased adaptive immunity, and more robust T cell-specific immunity with rapid antibody responses implicating the contribution of WT ORF8 in COVID-19 pathogenesis [58]. Further, no significant mutations are detected in ORF8 in omicron but D119- (deletion) in delta is implicated in impairing viral host immune responses resulting in increased viral pathogenesis [30]. The ω of *ORF8* is excessively increased in delta (ω = 5.48) than in alpha (ω = 0.00) and omicron (ω = 2.70) (Figure 5a, Table 1), indicating severe delta-specific pathogenesis including “cytokine storm”, and more COVID-19 hospitalizations compared to alpha- or omicron-mediated COVID-19. The membrane glycoprotein ORF5 (M) is involved in virus assembly and immune evasion [59]. The ω of *ORF5* reached its maximum in delta (ω = 0.85) but decreased in omicron (ω = 0.58) and alpha (ω = 0.0) (Figure 5b, Table 1), suggesting that its functional ability is superior to this virus assembly in delta than in the other two strains. In delta, the interaction of the ORF5 I82T mutation in C-terminal domain with ORF9 is implicated in elevated immune evasion [30]. The NSP10/NSP14/NSP16 complex forms a unique immune signature in COVID-19. NSP10 also induces IL8 through the NKRF pathway that manifests ARDs (Acute Respiratory Disorders) [60]. *NSP10* ω is increased maximally in delta (ω = 0.68) but decreased in alpha (ω = 0.00) and omicron (ω = 0.00) (Figure 5c, Table 1), indicating that some of the delta-specific severity could be attributed to NSP10 mutations. Further, no significant mutations are detected in NSP10 in omicron but D119-, T12I, T102I and A104V in delta stabilize the effect on binding NSP10 with NSP14 and NSP16 [61]. This binding stabilization is implicated in impairing viral host immune responses, resulting in increased viral pathogenesis [62]. NSP16 (2′-O-Ribose-Methyltransferase) methylates the 2′-hydroxy group of adenine and prevents the recognition of the viral RNA by host innate immunity [63]. In delta, the ω is increased (ω = 3.1) from alpha (ω = 0.09) and omicron (ω = 0.15) (Figure 5d, Table 1) signifying its contribution to the delta-specific pathogenesis and virulency. NSP5 (3C-like proteinase) promotes the degradation of IRF3, inhibits TGF-β kinases (MAP3K7), and favors NLRP12 inflammasome formation that induces multiple organ damage [19]. Although in alpha (ω = 0.00) and later in omicron (ω = 0.2) the *NSP5* ω is less, it is increased in delta (ω = 0.4) and early omicron (ω = 0.48) (Figure 5e, Table 1). A series of NSP5 mutations including G15S and K90R in delta suppressed IFNß production, allowing the virus to evade host immune responses [30]. Similarly, *ORF6* ω is maximally increased in delta (ω = 1.64) from alpha (ω = 0.75), and omicron (ω = 0.6) (Figure 5f, Table 1). ORF6 interacts with the TANK–TBK1–TRAF2 complex to modulate both the NFKB and IFN pathway to escape SARS-CoV-2 from host immune surveillance [60]. ORF6 promotes the overproduction and activation of Plasminogen Activator Inhibitor-1 (PAI-1), leading to coagulopathy that is increased in delta [64]. The most common mutation of *ORF6* in omicron is D61L, that showed increased IFN antagonism, whereas T21I and W27L in delta have increased pathogenicity and higher immune evasion with IFN antagonism [65]. The transmembrane protein ORF7A antagonizes IFN responses, and suppresses host immune activation [66]. The *ORF7A* ω is unchanged in alpha (ω = 0.0) and omicron (ω = 0.0) but highest in delta (ω = 2.6) (Figure 5g, Table 1), with an extensive divergent selection that contributes to the delta-specific severity of COVID-19. Although no significant nonsynonymous mutations are identified in omicron, three common mutations (P45L, V82A and T120I) in delta are associated with a severe and critical form of COVID-19 [67].

### 2.9. The ω of Several Genes Are Highest in Initial Omicron but Decreased in Subsequent Months

The ω of *spike* (*S*), *ORF9*, *ORF3A*, *ORF8*, *ORF5*, *NSP4*, *NSP9*, *NSP5*, *NSP10* and *NSP15* reached its maximum in earlier omicron in Jan 2022 but decreased in subsequent months (Table 1). These genes’ functional efficiency appears to be optimal in early omicron and then subsequently decreases. After the initial period, the omicron also showed reduced virulency in the pandemic.

Although the reason for the reduction in the ω in these genes in subsequent months is unknown, they could contribute to the weakening of the virus in later stages of omicron [68]. Nevertheless, the maximum values of ω are obtained in *spike*, *ORF8*, *NSP4* and *NSP15* in omicron, suggesting their maximum contribution in creating novel strains and potentially making them a target for drug development.

## 3. Discussion

Effective “vaccines” or medicines’ development against a virus requires a new mutation profile with evolutionary trajectories that could be replicated in vitro to predict the reversion of the vaccine effect, and its virulency as shown in other RNA viruses [69]. Twenty-nine [44] SARS-CoV-2 genes are primarily grouped as (1) entry, (2) replication and multiplication, (3) virus assembly, packaging and release and (4) host immune evasion (Appendix A). The role of each gene has been depicted from SARS-CoV-2 entry into human cells to the immune modulation to a path to virulency (Figure 6).

During the transition from bats, most of the SARS-CoV-2 proteins faced novel environmental challenges to function efficiently in the human system leading to an increased ω. As a result, numerous nonsynonymous mutations indeed shaped the evolution of virulent strains [70] (Appendix A).

Spike protein variations led to different levels of infective abilities in alpha, delta, and omicron [16]. Omicron accumulated maximum nonsynonymous mutations in the spike gene to create catastrophic effects in the pandemic [71] and showed the highest level of infecting ability, surpassing alpha and delta. Further, almost all SARS-CoV-2 vaccines are developed using the spike RBD (Receptor Binding Domain, 222 nt) RNA sequences, and the most accumulation of nonsynonymous mutations in RBD occurred in omicron (*n* = 15), compared to delta (*n* = 3), alpha (*n* = 1) or the original strain (*n* = 0) [72,73], that conferred structural variations in spike protein. A H69/V70 deletion in alpha has resulted in increased spike infectivity [74]. Alpha had almost 60% more infection ability than the originally emerged SARS-CoV-2 [24], and the delta variant had both increased infective ability with the most virulency [75,76]. The South African alpha showed ninefold less effectiveness for neutralizing the virus for the RNA vaccine of Pfizer [77]. The delta prolonged the pandemic by reducing the efficiency of currently used vaccines [16]. In delta, P681R mutation increases the replication fitness, leading to a higher viral load and transmissibility [78] and E156/R158 deletion conferred immune escape [79]. Omicron is less harmful than alpha or delta, although its vaccine neutralizing ability is more than the other two strains [14,68,71]. Omicron RBD mutations K417N, G446S, E484A, Q493R, G496S, Q498R and N501Y are associated with a decreased antibody binding [12] and the E484K, S377G/N/R and N439K are the most important mutations for immune evasion [13,80]. In addition to P681R, another mutation N679K in omicron showed combined effects to further increase the transmissibility [81,82]. Further, changes in K417N, N440K, G446S, S477N, T478K, E484A, Q493K, G496S, Q498R, N501Y, and Y505H in omicron showed an increased strength of RBD binding to ACE2 due to increased electrostatic interactions, new salt bridges and hydrogen bond formations than other SARS-CoV-2 strains, leading to its higher prevalence [12,82,83]. Thus, the constant increase in nonsynonymous mutations in spike RBD, leading to increased ω, supports the gradual increase in subsequent vaccine neutralizing efficiency from original strain to alpha, delta, and omicron [18]. Other entry-associated genes, NSP1 and NSP15, involved in infection, also show an increasing ω although the significance of individual contributions leading to an increased infection could not be determined.

The ω of replication and multiplication machinery genes of SARS-CoV-2 are continuously being increased to optimize their replication systems in the human host and have yet to be evolutionarily stabilized or saturated. The effect of this increased ω is evidenced by the increased replication and multiplication in later strains. It was observed that the omicron replicated about seventy times faster in bronchial tissue at 24 h after infection than the delta and the original strain [84]. The basic reproduction number (R_0_) is an indicator of the multiplication time that estimates the potentiality of an infected person to contribute to developing a pandemic, and R_0_ > 1 has less incubation time than R_0_ < 1. In Europe, alpha has 43–90% higher R_0_ than the original SARS-CoV-2 strain [76] and delta has increased R_0_ compared to alpha (Appendix A) [85,86]. In a study of Norway’s population, the first symptoms appeared within 5–6 days with the original strain, whereas a shorter incubation time is observed within 3 days after infection with omicron [37].

The immunomodulatory genes (Appendix A) show reduced ω in omicron and alpha but have a maximal peak in delta, leading to delta-specific severe symptomatic pathogenesis and hospitalization in COVID-19 compared to alpha and omicron. Alpha infection activated the major antiviral mechanism involving IFN induction and the JAK-STAT1 pathway, whereas genetic differences in delta suppressed IFN dependent STAT1 phosphorylation, leading to delta-specific immune vulnerability [87]. Omicron showed an insufficient or delayed IFN response without total suppression, which are attributed to having a milder phenotype than delta [88]. Delta-specific mutations in ORF8 (D119-), ORF5 (I82T), NSP10 (L133F), NSP16 (K160R and G77R), NSP5 (G15S and K90R), ORF6 (G15S and K90R), ORF7A (P45L, V82A and T120I) act as IFN antagonists and are associated with severe COVID-19 illness [30,61,65,67]. Delta also showed the highest peak of ω in some non-immune related genes (*ORF5*, *NSP14*, and *NSP16*) involved in viral assembly, packaging, and transport. A better virus packaging system than alpha or omicron also contributed to delta-mediated severity. A series of delta-specific mutations (ORF5, I82T; NSP14,I42V;NSP16, K160R and G77R) contributed to an increased vulnerability compared to alpha and omicron [30]. Mutations within *ORF9B* and *ORF6* genes leads to a large increase in sgRNA (subgenomic RNA) in alpha, showing a better antagonism of innate immune responses than the original strain [45]. Thus, the highest ω value of *ORF9* and *ORF6* is observed in delta, indicating further increased virulence over alpha and omicron.

Genes that have reduced or remained unchanged (*NSP6*, *ORF10*, *NSP2*) signify that these genes/proteins have optimal function in bats as well as in human hosts and do not need further modifications. It is also possible that these gene functions are important in bats for survival but not in humans. NSP6 and NSP2 are involved in autophagosome formation and lipid transport, respectively [30,52], and such molecular functions do not face immunological challenges from the host. In contrast, ORF10 is a very small protein (38 aa (amino acids)) that changes HLA epitope binding, and the process requires profound accuracy in such a way that the selection does not allow it to change, that is, in other words, mutations are abolished from the viral population [56].

Here the ω value for each gene has been validated by five established methods including “NG”, “ML” methods and other advanced methods, such as LPB93 [89,90,91]. Values of ω from all five methods are comparable with little differences. The ω value of each gene is correlated with experimental evidence and epidemiological studies that justifies the increase or decrease in ω. However, we did not address the importance of specific base changes leading to aa changes in a protein, and its effect on the particular protein function. Also, we did not include any insertion or deletion of bases during ω estimation, as PAML software does not allow such changes but takes only strings of codon as an input. The other three genes (*ORF3B*, 22 aa; *ORF3C*, 41 aa; *ORF3D*, 57 aa) are very small genes with few nt and few coding aa carrying genes that did not accumulate enough nonsynonymous mutations to affect a change in ω value, as is observed for *ORF10* (38 aa), and are not presented. Also, for ω estimation, we analyzed approximately over a hundred SARS-CoV-2 genomes for each group and for each strain, and that appears to be enough as fewer numbers of the genome (~50) does not markedly change their ω values [92].

Collectively, the trend of the increase in the ω in most of the replication and immunomodulatory genes predicts the emergence of novel and more virulent strains in future with better immune evasion power and super-infectivity. A maximal increase in ω in spike (*S*), *NSP4*, *NSP15* and *ORF8* in omicron has additional significance for infection, virulence and vaccine neutralizations. These genes could be further targeted for omicron-specific drug development. The temporal estimation of the gene-specific ω indicates an overall function of a gene in the evolutionary path of a virus and provides the basis for identifying genes for drug targets in COVID-19 transmission, severity, hospitalization, and vaccine neutralization. Nevertheless, the gene-specific estimation of ω could be used to dictate the power of infection, virulency and vaccine neutralizations in other emerging viruses.

## 4. Materials and Methods

### 4.1. Genomic Sequences

SARS-CoV-2 genomic sequences are obtained from the NIH SARS-CoV-2 repository (https://www.ncbi.nlm.nih.gov/nuccore/, accessed on 20 July 2022). This repository has information including isolation date, place/country, lineage and sequencing release dates. The accession numbers of each SARS-CoV-2 genome (29,303 bases) for each month carrying alpha, delta and omicron are listed (Appendix A). In each month group, SARS-CoV-2 genomes were collected from different geographical places in the world to maintain diversification. The sampling design is essentially random. All sequences from specific months belong to the same variant. This has been confirmed during mutation detection in NEXTCLADE (https://clades.nextstrain.org/, accessed on 10 February 2023) that showed the strain etiologies and lineages. The outliers were excluded from the group of strains of the same months. Over eight-hundred SARS-CoV-2 genomes (2.3 × 10^7^ bases) of these major strains are used in this study.

Although several strains emerged throughout the pandemic, here three major strains, alpha, delta and omicron, are compared with the original strain that emerged in December 2020. The alpha strain was first recognized in September 2020, whereas delta appeared in May–June 2021 (www.cdc.gov, accessed on 12 July 2022). Omicron, that started wave of infections, was originally recognized in December 2021–January 2022 and continued to be a major strain throughout 2022. This information is used to select the specific months to collect the SARS-CoV-2 genome for analysis.

### 4.2. Identification of Mutations

Each group consists of approximately over 100 genomes of SARS-CoV-2 for each month that includes a major strain, such as alpha (September 2020), delta (June and July 2021), or omicron (Jan, March, April, May 2022), etc. The whole genomic sequences of each month/strain are used as an input to run in NEXTCLADE (https://clades.nextstrain.org/, accessed on 12 December 2022) against the reference genome, Wuhan-Hu-1/2019 (MN908947). All mutations in each month group are listed in an excel file and filtered for unique mutations in that month by removing duplicates and multiple occurrences of a single base mutation. Large deletions, insertions and repeated mutations are excluded. All unique mutations of a month group for each strain are incorporated into the reference genome to construct a consensus for synthetic SARS-CoV-2 genome sequences that are representative of that month of the particular strain and used for ω analysis. The nucleotides (nt) of each synthetic genome for each strain are translated (https://www.expasy.ch, accessed on 25 September 2022) into amino acid (aa) sequences and all aa are removed, keeping only triplet codons as an input of the software. All reconstructed synthetic genome sequences are aligned with ClustalOmega (https://www.ebi.ac.uk/, accessed on 27 September 2022) and all nonsense codons are removed from all sequences as a software requirement.

### 4.3. Estimation of ω

Synonymous and nonsynonymous amino acid changes are estimated based on the “NG” method [89], the “ML” (Maximum Likelihood) method [90], Li et al. (1986)’s method [93], and other methods. These estimations were essentially conducted using the software PAML (phylogenetic Analysis by Maximum Likelihood, https://abacus.gene.ucl.ac.uk/software/paml.html, accessed on 27 September 2022). PAML independently calculates ω by all five methods. Nei and Gojobori (1986) estimation methods (ML—Maximum Likelihood) are based on the respective sum of all 1 nt and 2 nt (16 codons) changes of all synonymous (s_dj_) and nonsynonymous (n_dj_) codons as Sd (Σ^r^j = _1_sdj) and Nd (Σ^r^j = _1_ndj), and are determined for r number of j-th codon. The proportion of synonymous (P_s_) and nonsynonymous (P_n_) codons are determined by S_d_/S and N_d_/N, where S or N is the total expected alteration site in whole sequence length. The rate of synonymous (d_s_) and nonsynonymous (d_n_) changes are estimated by the formula d = −3/4Log e(1 − (4/3)p), where d is d_s_ or d_n_ and p is p_s_ or p_n_. d_n_/d_s_ is calculated as ω. Yang and Nelson (2000) used the “ML” method by incorporating the transition/transversion bias and unequal base frequencies by equal weighting in pathways that occur in mutation processes to calculate ω [90]. Li et al. and others developed three separate methods of estimating ω as LWL85 [93], LWL85m [94], and LPB93 [91], depending on nondegenerate, two-fold, and four-fold degenerate to a nucleotide site that estimates how often the site will result in aa replacement. Here, outputs of all methods are considered and used for graphs. Similar considerations are used for calculating each gene-specific ω, using an input sequence file consisting of specific gene sequences for each month of a particular strain.

As ω is an evolutionary measure, it needs at least two sequences to compute. Here all strain-specific ω represent the evolutionary changes from the original reference strain that emerged in December 2019 (MN908947).

### 4.4. Rate of Change in ω in Various Strains

The rate of ω is calculated in respect to time to generate mutations in each strain, and the ω of each strain are compared. That means the rate is a particular ω value/time taken to obtain this value in real time frame (e.g., for alpha, 9 months, and so on). So, the instant rate of ω for one month for alpha is 0.35 (the value of ω at Sept 2020)/9 (months) = 0.038 and the instant rate/site/year = 0.038 × 12 (months)/29,903 (genome size of SARS-CoV-2) = 1.561 × 10^−5^. Similarly, the average rate of ω in one month in alpha are ω_sept2020_ − ω_jan2020_/9 months = 0.35 − 0.04/9 = 0.034. The average rate of change in ω/site/year would be = 0.034 × 12/29,903 (genome size of SARS-CoV-2) = 1.382 × 10^−5^. The original values of ω have been taken from Table 1.

### 4.5. Phylogenetic Analysis

Phylogenetic analysis is performed in NEXTCLADE using a representative synthetic genome for each month of a particular strain. These synthetic genomes were used to build up the tree that would capture the distances among the consensus genome of each strain, containing the base changes in approximately a hundred SARS-CoV-2 genomes as a population in various parts of the world.

### 4.6. Statistical Analysis, Graphs, and Charts

This software instantly considers statistical parameters for estimating ω. All graphs are drawn in Excel. The value of ω (d_n_/d_s_) of −1 is replaced with 0 (zero) in all cases.

### 4.7. Visualization of Data and Rate of ω

The heatmap of ω is produced in R package (gplots, heatmap2.0). The rate of ω (dω/dt) is calculated in Excel. The values of ω in whole genome and spike gene were taken from Table 1. The average rate of change in ω_avg_ is (y2 − y1)/(x2 − x1) equivalent to (ω_sept2020_ − ω_emergence)_/(t_sept_ − t_emergence_) over the 29 months (January 2020 to May 2022) of time/month (t) of each strain. The instant rate of changes in ω in major strains are equivalent to the dω/dt over the continuous time (t).

## Figures and Tables

**Figure 1 ijms-25-06306-f001:**
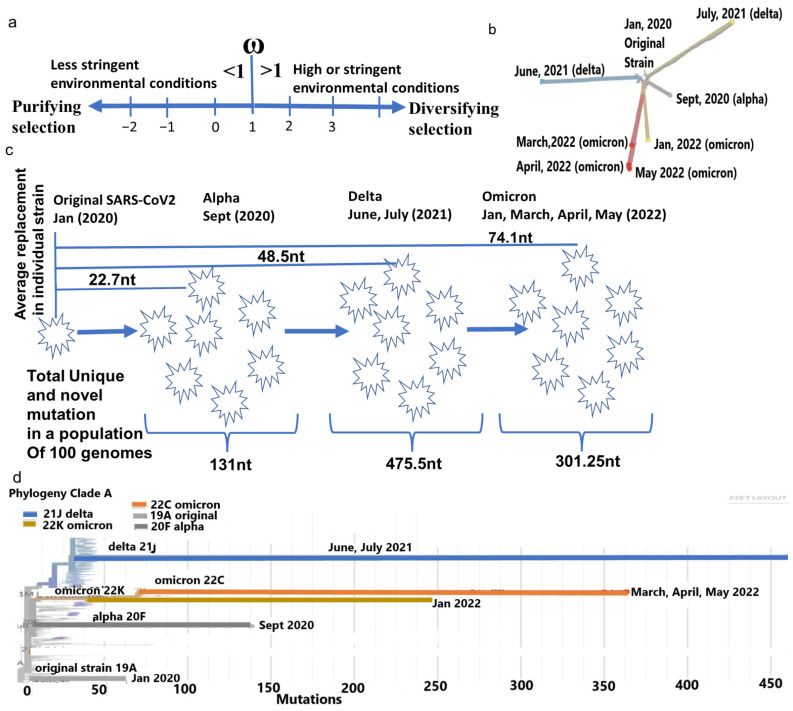
Phylogenetic differences of major SARS-CoV-2 strain. (**a**) Representation of the ω in respect to purifying selection in a favorable environment (<1) and diversifying selection (>1) in a more constrained environment. (**b**) The divergence of each major strain from the same ancestor. (**c**) Average mutation differences in a typical strain of alpha, delta, and omicron and population-specific mutation in each strain. (**d**) Omicron strains from April and May 2022 considerably differ genetically from January 2022 strains. Furthermore, all omicron strains have large genetic distances from the delta strain (June and July 2021).

**Figure 2 ijms-25-06306-f002:**
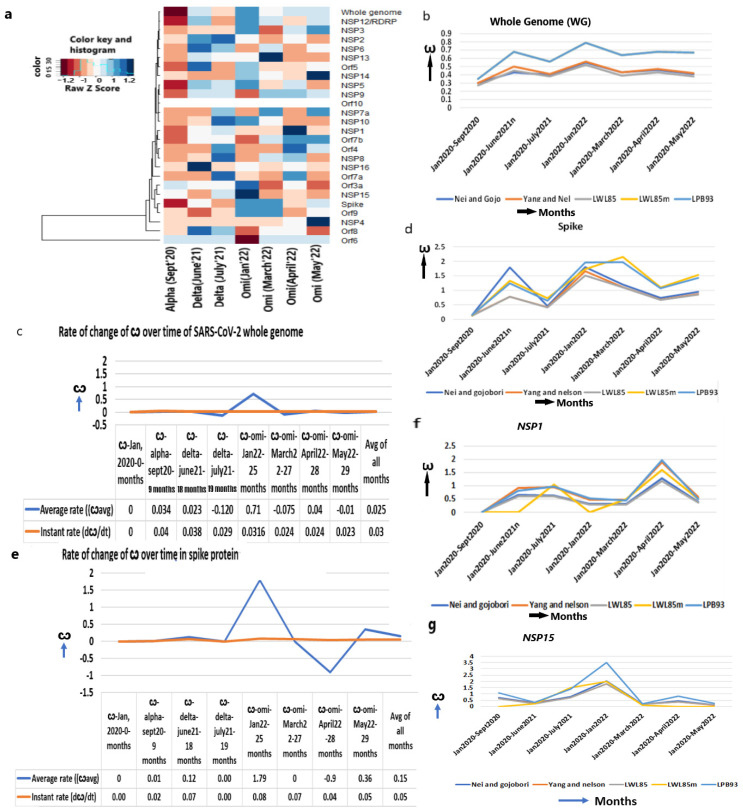
Increased ω in the whole genome and spike protein. (**a**) Heatmap of ω of whole genome and other genes from Jan 2020 to May 2022 in major strains shows mostly increased ω over the time period, (**b**) ω of whole SARS-CoV-2 genome, (**c**) The average and instant (dω/dt) rate of ω in whole genome, (**d**) ω of spike protein, (**e**) The average and instant (dω/dt) rate of ω in spike protein, (**f**) ω of *NSP1*, (**g**) ω of *NSP15*.

**Figure 3 ijms-25-06306-f003:**
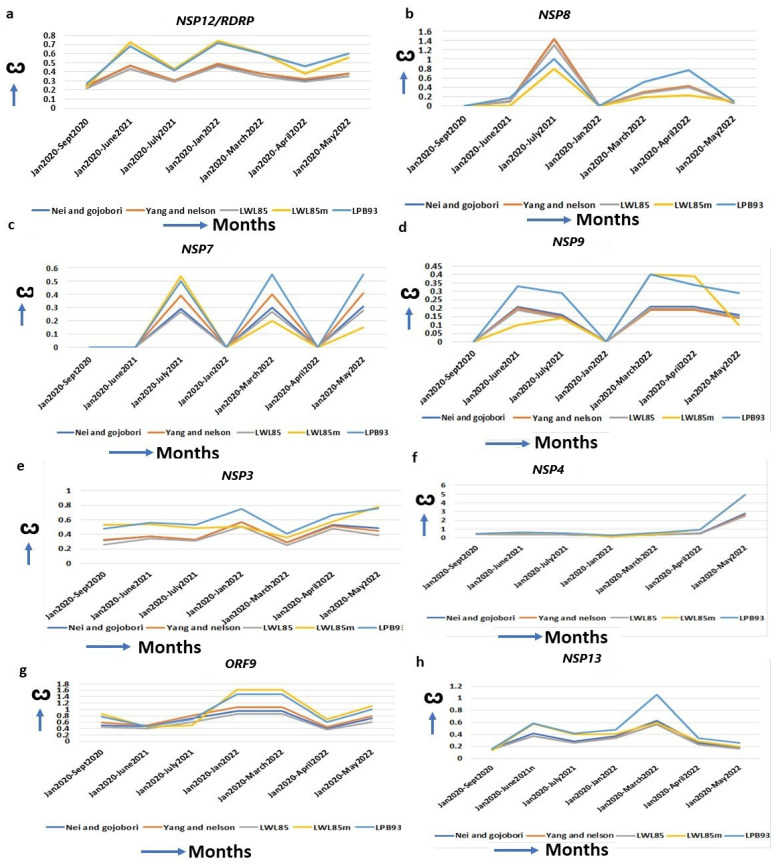
Increased ω for replication machinery-assisted genes. (**a**) *NSP12*, (**b**) *NSP8*, (**c**) *NSP7*, (**d**) *NSP9*, (**e**) *NSP3*, (**f**) *NSP4*, (**g**) *ORF9* and (**h**) *NSP13*. RDRP/NSP12 forms complexes with these proteins for viral replication, and the ω of these genes are increased continuously.

**Figure 4 ijms-25-06306-f004:**
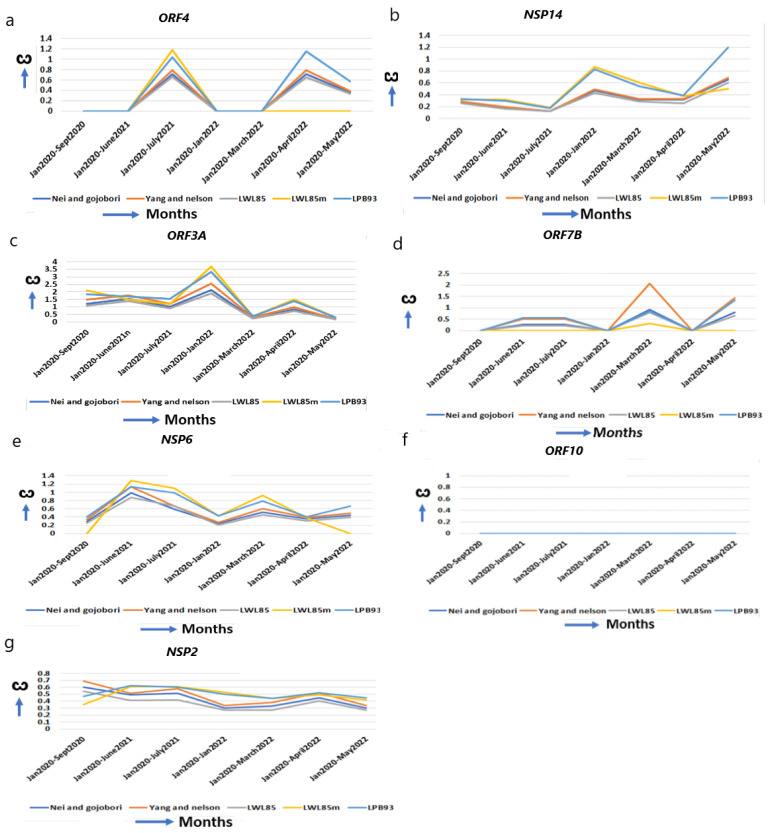
The ω of other functional genes and genes with unchanged or reduced ω. Except *NSP15* in Figure 2g, which helps in replication, (**a**,**b**) other genes help in virus entry, assembly, packaging, and virus release, (**a**) *ORF4*, (**b**) *NSP14*, (**c**) *ORF3A*, (**d**) *ORF7B*, (**e**–**g**) genes that have reduced or unchanged ω. (**e**) *NSP6* in omicron does not differ from alpha, (**f**) *ORF10*, no significant changes. (**g**) *NSP2* is reduced from the original strain.

**Figure 5 ijms-25-06306-f005:**
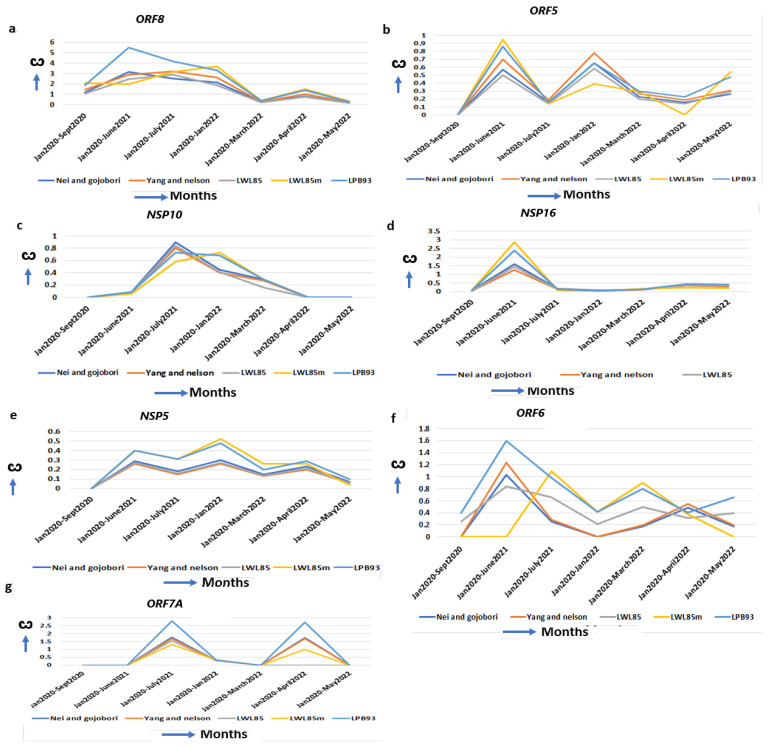
Delta-specific increase of ω and mechanism of virulence. (**a**) *ORF8*, (**b**) *ORF5*, (**c**) *NSP10*, (**d**) *NSP16*, (**e**) *NSP5*, (**f**) *ORF6* and (**g**) *ORF7A* also show the highest ω in delta strains implicating delta-specific virulency could be attributed to the optimal function of these genes.

**Figure 6 ijms-25-06306-f006:**
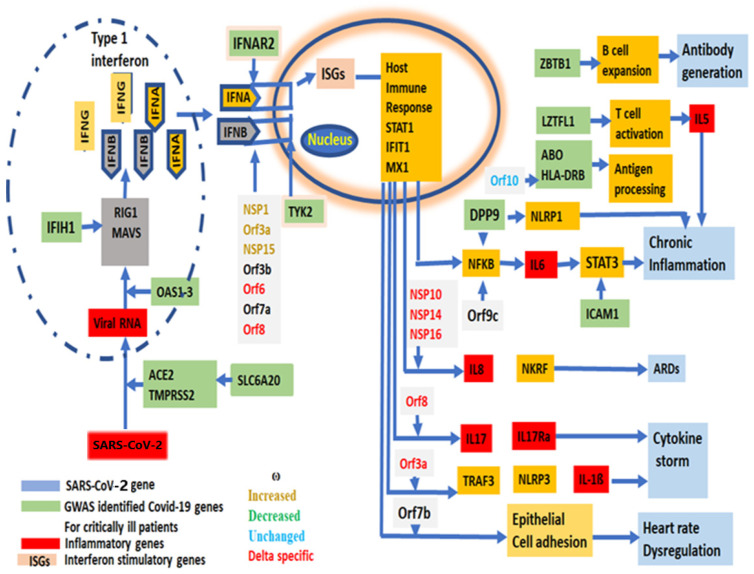
A schematic diagram of the virulency pathway of SARS-CoV-2 genes, showing the involvement of each SARS-CoV-2 gene in the stages from viral entry into human cells to the symptom development and virulency. GWAS identified genes associated with critically ill COVID-19 patients, showing their involvement in SARS-CoV-2 life cycle.

**Table 1 ijms-25-06306-t001:** ω variation in omicron strain. ω of each month determined by comparing the original strain, January 2020, using method LPB93, as it gave maximum ω value among the five methods in most of the genes.

Gene Name	ω-Alpha- Sept20	ω-Delta- June21	ω-Delta- July21	ω-Omi- Jan22	ω-Omi- March22	ω-Omi- April22	ω-Omi- May22
Whole Genome (WG)	0.35	0.68	0.56	0.79	0.64	0.68	0.67
*Spike*	0.15	1.24	0.64	1.97	1.97	1.07	1.43
*NSP1*	0	0.83	0.97	0.51	0.49	1.96	0.5
*NSP2*	0.47	0.6	0.62	0.5	0.44	0.51	0.45
*NSP3*	0.48	0.56	0.53	0.75	0.41	0.67	0.75
*NSP4*	0.49	0.63	0.54	0.33	0.57	0.94	4.94
*NSP5*	0	0.4	0.31	0.48	0.2	0.29	0.099
*NSP6*	0.4	1.13	0.98	0.43	0.79	0.41	0.66
*NSP7a*	0	0	0.5	0	0.54	0	0.55
*NSP8*	0	0.16	1	0	0.51	0.76	0.1
*NSP9*	0	0.33	0.29	0	0.39	0.34	0.29
*NSP10*	0	0.08	0.73	0.68	0.3	0	0
*NSP12/RDRP*	0.27	0.68	0.42	0.72	0.6	0.46	0.6
*NSP13*	0.17	0.59	0.42	0.48	1.06	0.24	0.26
*NSP14*	0.33	0.3	0.18	0.83	0.54	0.39	1.23
*NSP15*	1.07	0.34	1.4	3.56	0.2	0.81	0.27
*NSP16*	0.093	2.37	0.15	0.56	0.15	0.44	0.15
*ORF3A*	1.85	1.69	1.55	3.33	0.39	1.39	0.3
*ORF4*	0	0	1.04	0	0	1.15	0.57
*ORF5*	0	0.85	0.16	0.64	0.3	0.23	0.4
*ORF6*	0.4	1.64	0.98	−67.7	0.55	1.54	0.55
*ORF7aA*	0	0	2.8	0.33	0	2.76	0
*ORF7B*	0	0.56	0.56	0	0.84	1.29	1.31
*ORF8*	1.86	5.48	4.15	0.24	2.21	2.7	0
*ORF9*	0.79	0.47	0.7	1.47	1.47	0.6	1
*ORF10*	0	0	0	0	0	0	0

## Data Availability

All data are in the Appendix A provided with this manuscript.

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
