# Peer review of "Progressive Evolutionary Dynamics of Gene-Specific ω Led to the Emergence of Novel SARS-CoV-2 Strains Having Super-Infectivity and Virulence with Vaccine Neutralization"

_ijms, 2024, doi:10.3390/ijms25126306_

Round 1
Reviewer 1 Report
Comments and Suggestions for Authors
The author used bioinformatics methods to study the changes in ω across different strains at different times. While the logic is clear, the work lacks sufficient originality, and the conclusions reached are quite mundane. Although virus evolution is associated with changes in ω, the evolution is not driven by ω. Many issues have statistical correlations, but that does not prove physical causation, and often errors occur due to reverse causality. I find it difficult to agree with the author's description of changes in virulence during viral evolution. While the transmissibility of the virus is gradually increasing, the overall trend in virulence seems to be in the opposite direction. Additionally, the presentation of the data by the author is overly simplistic.
I personally do not favor the peer-review system, as I believe each paper should be subject to public scrutiny, and every piece of data, even the most mundane, could hold value in the future. However, in the case of this paper, I would suggest rejection. I do not appreciate lengthy discussions that merely list literature without adding any innovation. An article should have a central theme and present unique viewpoints in its discourse.
Author Response
Dear Reviewers,
Thanks for taking time to review this paper. I answered all concerns of the reviewers and modified the text wherever appropriate, based reviewer’s suggestion. All major changes are marked with “yellow” highlighter.
I hope it will be suitable for publication now.
Thanking you,
Amit K Maiti
Comments and Suggestions for Authors
Comments and Suggestions for Authors
Reviewer 1
Comments and Suggestions for Authors
The author used bioinformatics methods to study the changes in ω across different strains at different times. While the logic is clear, the work lacks sufficient originality, and the conclusions reached are quite mundane. Although virus evolution is associated with changes in ω, the evolution is not driven by ω. Many issues have statistical correlations, but that does not prove physical causation, and often errors occur due to reverse causality. I find it difficult to agree with the author's description of changes in virulence during viral evolution. While the transmissibility of the virus is gradually increasing, the overall trend in virulence seems to be in the opposite direction. Additionally, the presentation of the data by the author is overly simplistic.
Ans. Thanks for your comments. This work is an original work as it has been first estimation of ω of SARS-CoV-2. Moreover, this is the first estimation of strain specific comparison of ω through the evolutionary time/path of the virus. However, a direct conclusion has not been drawn except it has been shown how the SARS-CoV-2 evolution proceeded with the change of ω.
Also, there is misunderstanding about infection and virulence. In this study I also showed that the ω of infection causing genes are constantly increasing in and beyond omicron whereas the ω of virulent genes were highest in delta that was regarded as most virulent strain but decreased in omicron. For that reason, omicron lost virulency than delta.
I personally do not favor the peer-review system, as I believe each paper should be subject to public scrutiny, and every piece of data, even the most mundane, could hold value in the future. However, in the case of this paper, I would suggest rejection. I do not appreciate lengthy discussions that merely list literature without adding any innovation. An article should have a central theme and present unique viewpoints in its discourse.
Ans. Thanks for your message. Here the ω of each gene of three strains were estimated and epidemiological evidence and consequences of mutations of each increase/decrease of ω were presented to justify their increase or decrease. For that the discussion has been increased in length. Without explaining and supporting evidence, it would have been incomplete for increase/decrease of ω in each case.
Reviewer 2 Report
Comments and Suggestions for Authors
1) Lines 25-27, this sentences seems to come out of nowhere, as if the viral evolution were happening on its won (when it's not). Please provide more context for this sentence.
2)The introduction fails to consider the ongoing transmission in multiple regions of the world, the heterogeneity of measures to reduce its spread, the wide availability of vaccines, and the different vaccination schedules employed, as well as the interconnectedness of societies and globalization.
3) Please, prior results provide a deeper explanation about omega concept and value. Is not clear their significance and impact.
4) In figure 2b to 2f, authors show puntual data of omega, but this results do not have any confidence interval? How we can be sure about that. Please provide dispersion data of omega for figure 2. Also with results from line 106-110. Te authors provide a value without data distribution. Therefore we cannot know if the range is wide of narrow.
5) In the same idea, how the authors can wrote about a higher rate, without intervals? Lines 111-133.
7) Same comment to table 1, figure 3 to 5. The authors only show a puntal data without dispersion data.
8) Figure 6 is not related with the results and with the main idea of the paper.
Author Response
Dear Reviewers,
Thanks for taking time to review this paper. I answered all concerns of the reviewers and modified the text wherever appropriate, based reviewer’s suggestion. All major changes are marked with “yellow” highlighter.
I hope it will be suitable for publication now.
Thanking you,
Amit K Maiti
Reviewer 2
- Lines 25-27, this sentences seems to come out of nowhere, as if the viral evolution were happening on its won (when it's not). Please provide more context for this sentence.
Ans. Thanks for your message. I expanded this with addition of sentences (line 26-30) in the context.in the revised manuscript.
2)The introduction fails to consider the ongoing transmission in multiple regions of the world, the heterogeneity of measures to reduce its spread, the wide availability of vaccines, and the different vaccination schedules employed, as well as the interconnectedness of societies and globalization.
Ans. Thanks for your message. I added the heterogenous measure of spread and vaccine consequences in the introduction in the revised manuscript.
3)Please, prior results provide a deeper explanation about omega concept and value. Is not clear their significance and impact.
Ans. Thanks for your message. I explained ω and its significances with impact in virus evolution explosively in introduction. However, I explained it deeper in results prior to ω section in the revised sec manuscript.
4)In figure 2b to 2f, authors show puntual data of omega, but these results do not have any confidence interval? How we can be sure about that. Please provide dispersion data of omega for figure 2. Also, with results from line 106-110. The authors provide a value without data distribution. Therefore, we cannot know if the range is wide or narrow.
Ans. Thanks for your comments. I think there is a misunderstanding about ‘puntual data”. I don’t understand this, also tried to look here and there, but did not find what is meant by “ punctual data”?
The value of ω is PAML software outputs that calculates the number of nonsynonymous mutations to synonymous mutations in a protein sequence derived from DNA sequence and interpret with codon degeneracy that gives weightage to each base position in a triplet codon. The software itself considers statistical corrections for possibility and dn/ds is a complicated output. In this study, five methods (that’s the number of methods exists until now) of ω estimation was calculated and presented each one in these graphs for comparison to validate that all methods are giving same trend (with very minute differences) for each strain. Although the value of all methods is not presented in the tabular form but included in all graphs as separate methods. Also, I did not calculate any average of five methods or any other probability to show which method is correct or diverted from the rest. Comparison of various methods is not the purpose of this paper and that had been done when people published their new methods, and they already compared them with statistical analysis. However, as the trend is similar in the graph (The value of only one method is presented in the table-the highest values-mentioned in the table legend), it is not needed to get average value and conclusion. For that reason, there is no distribution or calculation of confidence interval or p-value.
There is a trend of performing one method, but I estimated ω in all methods and presented in the graph to show how they gave similar results.
- In the same idea, how the authors can write about a higher rate, without intervals? Lines 111-133.
Ans. Thanks for your message. The rate of ω is calculated in respect to time to generate mutation in each strain and the ω of each strain are compared. That means, the rate is in particular ω value/time taken to obtain this value in real time frame (e.g.for alpha, 9 months, for delta, 18 months and so on). So, the instant rate of ω for one month for alpha is 0.35 (the value of ω at sept, 2020) /9(months)=0.04 and the instant rate/site/year=0.04*12 (months) *29903 genome size of SARS-CoV-2)=1.561E-05. Similarly, the average rate of ω in one month in alpha are ωsept2020- ωjan2020/9 months= 0.35-0.04/9=.034. The average rate of change in ω/site/year would be =.034*12/29903(genome size of SARS-CoV2-))=1.382E-05. The original values of ω have been taken from Table 1. I explained a separate paragraph of determination of rate of ω in the materials and methods in the revised manuscript.
7) Same comment to table 1, figure 3 to 5. The authors only show a puntal data without dispersion data.
Ans. Thanks for your message. Again, I don’t understand what is a (“puntal data”)? All figures have the values of ω of all methods but table 1 only present a method that gave the highest value. Each method gives a single value for ω and there is no distribution. It is mathematical calculation of numbers of mutations adjusted with codon degeneracy.
8) Figure 6 is not related with the results and with the main idea of the paper.
Ans. Thanks for your message. I agree with the reviewer, this figure does not align with the main idea of the paper. But it is kept to explain ω of which genes in the SARs-CoV-2 life cycle is increased or decreased that contributed in the infection and virulence.
Reviewer 3 Report
Comments and Suggestions for Authors
Maiti analyzed the proportion of nonsynonymous to synonymous mutation (dn/ds,êž·) of the SARS-CoV-2 genome and showed that marked increase of the êž· occurred in spike gene from alpha (êž·=0.2) to omicron (êž·=1.97). the author also showed that the replication machinery genes, such as RDRP, NSP3, NSP4, NSP7, NSP8, NSP10, NSP13, NSP14, and ORF9 are yet to be evolutionary stabilized and contributing to the evolution of novel virulent strains. The topic is important and interesting. The study on the êž· of SARS-CoV-2 whole genome and its’ genes can provide new insights into virus virulence, vaccine-induced antibody neutralization and infectivity. Please see the suggestions below.
1. The author may need several sentences in the discussion section to show the decrease of êž· in genes and what are the possible reasons.
2. Several sentences about the importance of the study are needed in introduction section.
3. Line 128, please be specific about “environmental niches”.
4. Line 111, the increase of the êž· of spike gene increases infection. Does increase of the êž· in other genes may also contribute to the infection?
Author Response
Dear Reviewers,
Thanks for taking time to review this paper. I answered all concerns of the reviewers and modified the text wherever appropriate, based reviewer’s suggestion. All major changes are marked with “yellow” highlighter.
I hope it will be suitable for publication now.
Thanking you,
Amit K Maiti
Reviewer 3
Comments and Suggestions for Authors
Maiti analyzed the proportion of nonsynonymous to synonymous mutation (dn/ds,êž·) of the SARS-CoV-2 genome and showed that marked increase of the êž· occurred in spike gene from alpha (êž·=0.2) to omicron (êž·=1.97). The author also showed that the replication machinery genes, such as RDRP, NSP3, NSP4, NSP7, NSP8, NSP10, NSP13, NSP14, and ORF9 are yet to be evolutionary stabilized and contributing to the evolution of novel virulent strains. The topic is important and interesting. The study on the êž· of SARS-CoV-2 whole genome and its’ genes can provide new insights into virus virulence, vaccine-induced antibody neutralization and infectivity.
Please see the suggestions below.
- The author may need several sentences in the discussion section to show the decrease of êž· in genes and what are the possible reasons.
Ans. Thanks for your message. I explained additional reasons and consequences for genes with decreased ω.
- Several sentences about the importance of the study are needed in the introduction section.
Ans. Thanks for your message. I added text in the introduction for importance of the study.
- Line 128, please be specific about “environmental niches”.
Ans. Thanks for your message. I explained environmental niches in the revised manuscript.
- Line 111, the increase of the êž· of spike gene increases infection. Does increase of the êž· in other genes may also contribute to the infection?
Ans. Thanks for your message. Other infection related genes (entry associated genes, NSP1 and NSP15) also have constantly increasing ω. However, it is not possible to conclude at this stage, how much increase of ω of any of these genes contributed how much increase of infection. Epidemiological evidence of mutations in each of these genes are correlated with increase of infection (several examples are in the text). However, I discussed this in the discussion in the revised manuscript.
Round 2
Reviewer 2 Report
Comments and Suggestions for Authors
The corrections improved the manuscript
Reviewer 3 Report
Comments and Suggestions for Authors
this is an updated version, and the author has taken suggestions into consideration, no concerns detected in this version.